# Axon-like active signal transmission

Timothy D. Brown[1], Alan Zhang[1], Frederick U. Nitta[1,2], Elliot D. Grant[1], Jenny L. Chong[3], Jacklyn Zhu[1], Sritharini Radhakrishnan[1], Mahnaz Islam[1,2], Elliot J. Fuller[1], A. Alec Talin[1], Patrick J. Shamberger[3], Eric Pop[2], R. Stanley Williams[1,3] & Suhas Kumar[1✉]

Any electrical signal propagating in a metallic conductor loses amplitude due to the natural resistance of the metal. Compensating for such losses presently requires repeatedly breaking the conductor and interposing amplifiers that consume and regenerate the signal. This century-old primitive severely constrains the design and performance of modern interconnect-dense chips[1]. Here we present a fundamentally different primitive based on semi-stable edge of chaos (EOC)[2,3], a long-theorized but experimentally elusive regime that underlies active (self-amplifying) transmission in biological axons[4,5]. By electrically accessing the spin crossover in LaCoO$_3$, we isolate semi-stable EOC, characterized by small-signal negative resistance and amplification of perturbations[6,7]. In a metallic line atop a medium biased at EOC, a signal input at one end exits the other end amplified, without passing through a separate amplifying component. While superficially resembling superconductivity, active transmission offers controllably amplified time-varying small-signal propagation at normal temperature and pressure, but requires an electrically energized EOC medium. Operando thermal mapping reveals the mechanism of amplification—bias energy of the EOC medium, instead of fully dissipating as heat, is partly used to amplify signals in the metallic line, thereby enabling spatially continuous active transmission, which could transform the design and performance of complex electronic chips.

Because modern electronic chips strive to pack more components per unit area, interconnects are becoming longer, narrower, lossy and more densely packed[1,8,9]. Nearly a decade ago, the delays associated with resistive–capacitive losses across interconnects exceeded the latencies of transistor-based switches[10]. In prevailing solutions to address the increased losses and delays, transmission lines are broken into smaller lengths, with carefully designed distributed layouts of complementary metal–oxide–semiconductor buffers and drivers interposed in the signal path[8] (Fig. 1a). Research trends, especially those in post-silicon technologies, point towards new low-loss interconnect materials, such as carbon nanotubes and two-dimensional materials separated by insulators with low dielectric constants[11–14]. However, these solutions still require transmission to be spatially broken to accommodate amplifiers, buffers and repeaters, leading to restrictive, time-consuming and expensive chip design, as well as the degradation of chip-level performance.

Biological systems provide an excellent evolutionary blueprint that addresses this issue in a radically different way. Axons in neurons can transmit constant-amplitude action potentials across many metres of highly resistive organic material in large animals by using spatially distributed local amplification (that is, self-amplifying or active transmission, without needing separate amplifying components)[5,15] (Fig. 1b). The Hodgkin–Huxley neuron models developed in the 1940s and 1950s revealed that the propagation of nerve impulses in axons used recurring amplification in the nodes of Ranvier—gaps in the insulating myelin sheath—through coordinated sodium and potassium channels[16,17]. Models by FitzHugh and Nagumo in the 1960s further demonstrated

that amplification in axons was associated with small-signal negative resistances[18,19] (Fig. 1c), which were previously known to amplify audio signals[20,21]. In 2012, a mathematical connection between the complex neuronal action potential, small-signal negative resistance and EOC was established[4,6,22]. EOC, a principle introduced in the late 1980s, is locally active (making it capable of amplification of fluctuations) as well as locally stable (preventing small fluctuations from growing without bounds)[2,23,24]. EOC has since been used to describe many phenomena in nonlinear cellular media and biological dynamics[25,26]. This principle, although predicted decades ago, has remained elusive to experimental isolation.

In this study, we used LaCoO$_3$, a spin crossover material that undergoes a gradual change from low- to high-spin states accompanied by a decrease in resistance with increasing temperature[27,28]. By electrically accessing the spin crossover in a two-terminal LaCoO$_3$ test device, we experimentally verified the criteria for semi-stable EOC: a non-oscillating negative differential resistance (NDR), a magnitude of reactive phase shift between the current and voltage possible only at semi-stable EOC and controlled amplification of current or voltage fluctuations. Next, we assembled a metallic transmission line of length 1 mm on the LaCoO$_3$ film, and showed that when the underlying medium is held at semi-stable EOC using a direct-current (d.c.) bias, a time-varying alternating-current (a.c.) signal entering the transmission line on one end exits the other end amplified (Fig. 1d). Thus, an entire metallic transmission line was locally activated and its internal resistance was fully overcome to enable self-amplifying active signal transmission, without any spatial discontinuities as in existing primitives. Finally, operando

[1]Sandia National Laboratories, Livermore, CA, USA. [2]Department of Electrical Engineering, Stanford University, Stanford, CA, USA. [3]Texas A&M University, College Station, TX, USA. ✉e-mail: su1@alumni.stanford.edu

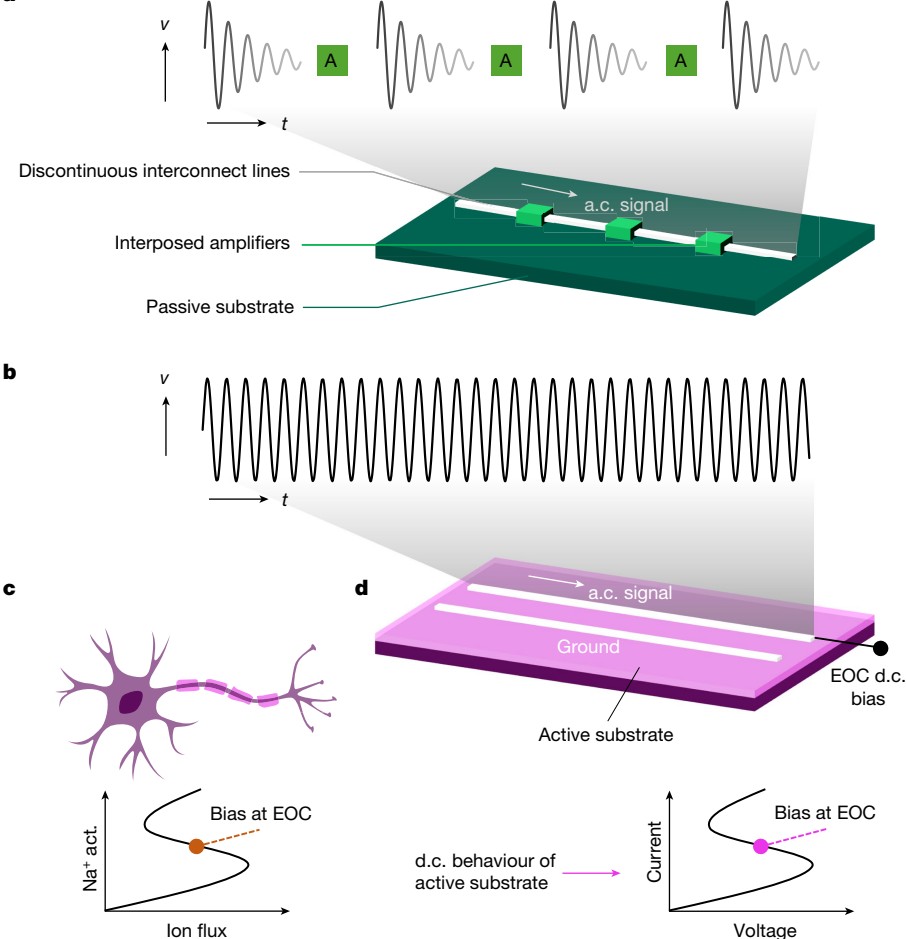

**Fig. 1 | Bio-inspired active transmission. a**, Prevailing solution to on-chip signal transmission, in which the signal path is broken by repeaters or amplifiers (A), which re-amplify the decaying signal. **b**, The bio-inspired transmission primitive in which signal amplification occurs continuously throughout an unbroken signal path. **c**, Neurons operate in a region of non-monotonic responses between the ion flux and Na$^+$ ion activation (Na$^+$ act.) channels. Such a behaviour, similar to an NDR, is theoretically predicted to embody semi-stable EOC, which enables local amplification. **d**, Electronic version of a bio-inspired active-medium transmission, based on an active medium that exhibits NDR and EOC.

thermal mapping revealed that relative to the d.c. Joule heating with no a.c. signal applied (consequently, no signal amplification), the d.c. Joule heating was reduced during the amplification of an a.c. signal. That is, when electrically biased at semi-stable EOC, some portion of energy that is dissipated as heat in most electronic components can be made to amplify small signals carried by an embedded metallic line.

## Edge of chaos criteria

We recently mapped the mathematical theory of local activity onto a physical electrothermal volatile switch with temperature as a single state variable[7,29,30]. The analysis indicated that semi-stable EOC is a special case of local activity—the ability to amplify small fluctuations. Although all other instances of local activity are unstable with respect to all fluctuations, semi-stable EOC is conditionally stable. That is, a resistive component operating at semi-stable EOC is unstable in response to either current or voltage fluctuations, but not both. This semi-stability leads to three clear experimental manifestations, which serve as required conditions to verify the presence of semi-stable EOC.

First, specifically in electrothermal memristors, for which conductance increases with the temperature state variable, semi-stable EOC allows the accessibility of non-oscillating current-controlled NDR (voltage decreases as the current is increased) by a steady-state current bias (but not a voltage bias). For completeness, this statement has rare and

subtle exceptions, outlined elsewhere[30]. Symmetrically, a region of positive differential resistance (PDR) lacks the necessary qualities of semi-stability and amplification. Second, the semi-stability at EOC manifests as a stable pole and an unstable zero in the Laplace transfer function that relates the small-signal current and voltage (Supplementary Note 1). Thus, in a certain bandwidth, the phase shift between current and voltage a.c. fluctuations $\Delta\phi$ (ranging from $-\pi$ to $\pi$) lies in the interval $\frac{\pi}{2} < |\Delta\phi| \le \pi$. This phase shift exceeds the maximum possible magnitude ($\pm\frac{\pi}{2}$) for any combination of passive electrical components. Here, we denote phase shifts in the range $\frac{\pi}{2} < |\Delta\phi| \le \pi$ as super-quadrature, whereas those with $|\Delta\phi| \le \frac{\pi}{2}$ are sub-quadrature. Third, the semi-stability at EOC implies that small-signal power fluctuations neither decay nor grow without bound; instead, they are controllably amplified. Thus, there is a finite bandwidth where a bias in the NDR region leads to both super-quadrature phase shifts and small-signal a.c. power amplification.

If a medium biased at semi-stable EOC is coupled to a destabilizing element, such as a capacitor, the combined system produces self-sustained neuronal dynamics, such as periodic oscillations, action potentials, aperiodic spiking, bursting and chaos[31–34]. Most efforts examining such behaviours have used abrupt electronic conductance changes, such as first-order phase transitions (especially the Mott transition in VO$_2$ and NbO$_2$), which are difficult to stabilize[35–42]. Such self-sustained dynamics, although useful in many applications,

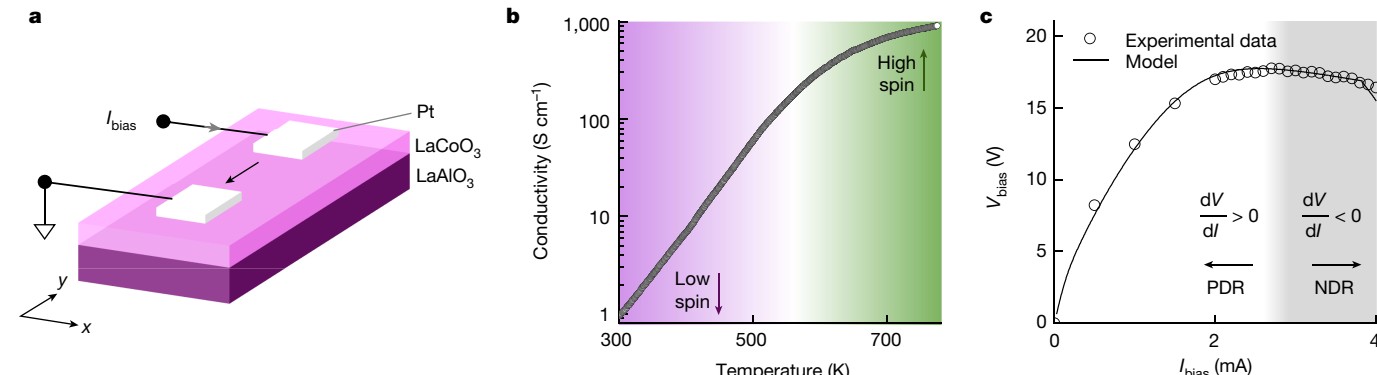

**Fig. 2 | Test structure, material and its quasistatic electrical behaviour.**
**a**, Schematic of the test structure. The phase-shift and noise measurements were performed on components with electrode widths (in the $x$ direction) of 100 μm, with a gap between the electrodes (in the $y$ direction) of 6 μm.

**b**, Temperature-dependent electrical conductivity of $LaCoO_3$ exhibiting a $10^3$ times nonlinear increase between 300 and 650 K. **c**, Measured quasistatic bias voltage ($V_{bias}$) as a function of applied bias current ($I_{bias}$), with no time-varying signals applied. The regions of NDR and PDR are marked.

make it difficult to isolate the underlying semi-stable EOC in the active medium. In the case of semi-stable EOC without a destabilizing influence, fluctuations are amplified without growing to the point where external factors dominate and drive self-sustained dynamics, a principle that has remained elusive thus far. Thus, to achieve axon-like active transmission, we first isolated semi-stable EOC with an appropriate material.

## Isolating the EOC

We constructed two-terminal test structures atop epitaxially grown $LaCoO_3$ (Fig. 2a), which were used to measure the gradual increase in the conductivity of $LaCoO_3$ (by roughly three orders of magnitude over a temperature range of 300 K; Fig. 2b), a known signature of its spin crossover[27,28]. On sourcing quasistatic currents ($I_{bias}$) and measuring the resulting voltage ($V_{bias}$), we observed a region of non-oscillating current-controlled NDR beginning at roughly 3 mA, in which the voltage was gradually reduced as the current was increased (Fig. 2c). This observation of current-controlled NDR in $LaCoO_3$ devices was robust with respect to both device dimension and between consecutive cycles on the same device. The Methods provides the film growth, device fabrication, measurement and analysis details. Thus, we verified the first criterion for semi-stable EOC.

To test the second and third criteria of semi-stable EOC, we used a function generator and oscilloscope to measure the phase shifts and amplitude gains of electrical signals on the compact device, but this time, taking the small-signal a.c. current through the device ($i(t)$) as the input and the a.c. voltage ($v(t)$) as the output for a given d.c. bias ($I_{bias}$) (Fig. 3a). To create a combined d.c. + a.c. current source with sufficient accuracy, we coupled the same power supply of the source measure unit used for the quasistatic current–voltage ($I$–$V$) measurements to the function generator using a diode and resistors (Methods). At a small d.c. bias current of 1 mA, in the PDR region, $\Delta\phi$ (phase difference between $i(t)$ and $v(t)$) was close to 0 at 100 Hz (Fig. 3b,c). The observation of $\Delta\phi \approx 0$ at low frequencies and at PDR biases confirmed that the extraneous reactance offered by the circuit (outside the component under study) was negligible at low frequencies. As we increased the d.c. current bias to the cusp of the PDR–NDR transition (roughly 3 mA), the $\Delta\phi$ value at 100 Hz increased smoothly and monotonically from 0 to $\frac{\pi}{2}$. Following the transition into the NDR region (a bias of roughly 3 mA), super-quadrature phase shifts were observed at low frequencies. At even higher biases (close to 4 mA), $\Delta\phi$ at 100 Hz was within 3% of π—the theoretical maximum. These phase-shift observations verify the second criterion of semi-stable EOC. We performed multiple checks throughout the experiment,

including increasing and reducing the d.c. bias, and confirmed the same measured phase shifts independent of the order in which data were collected. At higher frequencies, irrespective of the d.c. bias point (whether in NDR or PDR), the phase shifts converged to $-\frac{\pi}{2}$, an isolation of the inherent capacitive reactance of the circuit. This observation is expected as super-quadrature phase shifts in semi-stable EOC are bandwidth limited.

The same small-signal sinusoidal data used to study the phase shifts exhibit the third criterion for semi-stable EOC, namely, the amplification of fluctuations. To quantify the gain of fluctuation amplification, we measured the power spectral densities (PSDs) of the output signals corresponding to the 100 Hz sinusoidal input, normalized by the output power at 100 Hz (Fig. 3d). This normalization process decouples the changes in the amplitude of voltage fluctuation arising from the amplification due to semi-stable EOC under study, from the routinely expected voltage scaling due to the changing device resistance at different current biases (Methods). For d.c. biases around the onset of NDR, the normalized power increased across all the measured frequency components (up to roughly 10 kHz) by nearly two orders of magnitude (a difference of 20 dB). This increased magnitude of fluctuations is apparent in Fig. 3b, where the voltage response at a d.c. bias of 3 mA (at the onset of NDR) clearly contains amplified perturbations relative to a PDR bias of 1 mA (see Methods for the quantitative details). With the observation of amplified perturbations, all the three criteria were satisfied. Thus, we have isolated semi-stable EOC in $LaCoO_3$.

Interestingly, in Fig. 3d, the magnitude of amplified perturbations is maximized at a bias close to the onset of NDR (roughly 2.9–3.4 mA). There have also been previous reports on NDR-driven self-sustained oscillations that appeared irregular and chaotic when biased at the onset of NDR[43]. From our data, it is apparent that the underlying semi-stable EOC is the most susceptible to perturbations when the system exhibits the maximum magnitude of nonlinearity in the $I$–$V$ curve (that is, where the differential conductance $\frac{dI}{dV}$ diverges).

By using the theory of local activity reconstructed for electrothermal systems, we developed a quantitative compact model that accurately reproduced the quasistatic measurements (Fig. 2c, solid line), verifying that the nonlinear electrothermal transport and thermal dynamics caused the NDR. We further used the electrothermal compact model to reproduce the super-quadrature phase shifts at semi-stable EOC bias, in a typical error of 6% (Supplementary Note 2). Therefore, the super-quadrature phase shift of $LaCO_3$ is a simple but nearly unknown result of semi-stable EOC for nonlinear dynamic systems and small-signal sinusoidal excitations, which can be measured and easily reproduced using numerical calculations.

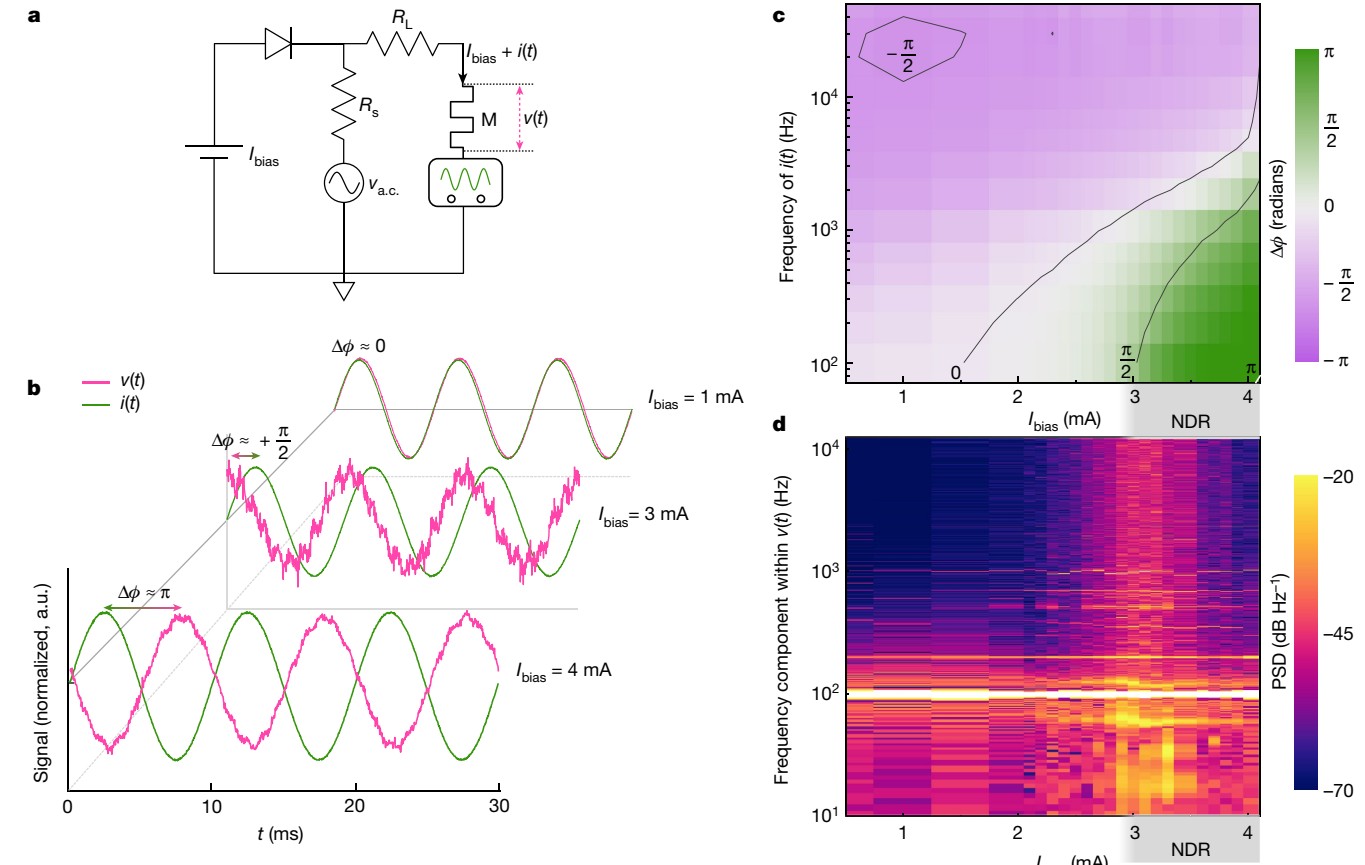

**Fig. 3 | Dynamic measurements for circuit, phase shifts and noise amplification. a**, Schematic of the circuit of the dynamic measurements, which featured the simultaneous application of a d.c. bias current and a time-varying small-signal current ($i(t)$), as well as a measurement of the resulting d.c. voltage and the time-varying small-signal voltage ($v(t)$). Resistor $R_s$ was used to set the a.c. small-signal amplitude, whereas $R_L$ suppressed self-oscillations at all the d.c. biases of the device, M, under study (Methods). **b**, Time evolution of $v(t)$ and $i(t)$ at different d.c. current biases with the frequency of $v(t)$ set to 100 Hz. The current biases correspond to PDR (1 mA), the cusp of the PDR–NDR transition (3 mA) and NDR (4 mA). The phase

shifts $\Delta\phi$ are marked, and $v(t)$ and $i(t)$ are scaled and plotted on the same ordinate to allow a visual comparison of their relative phases. The amplitude of $v_{a.c.}(t)$ in all the measurements was fixed (Methods). **c**, Contour plot of $\Delta\phi$ (corresponding to the colour) as a function of frequency and $I_{bias}$. Contour lines for $\Delta\phi$ of $-\frac{\pi}{2}$, 0, $\frac{\pi}{2}$ and $\pi$ are displayed. **d**, Contour plot of the PSD of the measured $\tilde{v}(t)$ (corresponding to the colour) as a function of frequency and $I_{bias}$. The yellow band of increased PSD across all the frequencies is apparent at $I_{bias}$ around 2.9–3.4 mA. All data were normalized to the data at 100 Hz. Data in **c** and **d** were obtained on the same device for which the quasistatic behaviour is displayed in Fig. 2c.

## Self-amplifying active transmission line

Having isolated semi-stable EOC, we next invoked the theoretically predicted link between semi-stable EOC and active (amplified) signal transmission in the axon[4]. We hypothesized that active transmission occurs in a transmission-line-like spatially extended device manifesting semi-stable EOC (Fig. 4a). Our transmission-line structure clamps the a.c. signal at one end of the line, and leaves the other end effectively floating. This allows the input signals to grow or decay at the floating end, as dictated by the resistance of the line itself and nonlinear dynamics of the system. Because the semi-stable EOC medium was capable of amplifying perturbations (in the test structures), the floating end of the transmission line (which interfaces with the semi-stable EOC medium) should output signals that are amplified compared with the input over some range of frequencies.

We tested our hypothesis by fabricating the simplest structure that could provide a proof of principle: two parallel metallic lines atop a thin film of epitaxial LaCoO$_3$ (Fig. 4a), each with a small bias (0.1 V) resistance of 6.6 kΩ mm$^{-1}$. Across many devices, the metallic lines ranged in length from 0.2 to 4 mm, representative of typical chip-layout dimensions. We used the metallic lines as electrodes to d.c. bias the underlying LaCoO$_3$, such that the bias power flowed from one line (at a specific d.c. current) to the other (held at electrical ground), through the LaCoO$_3$. On the line

held above ground, we used a function generator to input a small sinusoidal a.c. voltage signal at one end of the line, and measured the a.c. voltage signal exiting the same line at its other end with an oscilloscope. Thus, the a.c. signal was transmitted in the resistive metallic line perpendicular to the direction of the d.c. bias, not through the medium under bias. In the test structures (Figs. 2 and 3), we measured small-signal voltage outputs relative to small-signal current inputs, which allowed the quantification of $I$–$V$ phases. However, in the transmission lines, to measure gain across a common quantity, we measured the small-signal voltage output ($v_{out}(t)$) relative to the small-signal voltage inputs ($v_{in}(t)$). Gain was defined as the amplitude of $v_{out}(t)$ divided by the amplitude of $v_{in}(t)$.

When the underlying LaCoO$_3$ was biased at PDR, the signal from the 1-mm-long transmission line, measured using a signal of frequency 100 Hz, was attenuated (gain was below 0.8), as expected for a resistor (Fig. 4b). However, when we repeated the experiment with the underlying LaCoO$_3$ biased at NDR, the voltage at the output of the metallic transmission line was larger than the input (gain = 1.024 ± 0.007).

We also measured the signal at different points along the same transmission line (different distances from the input) (Fig. 4c). At 0.75 mm from the input, in NDR bias, we measured the maximum gain of 1.639 ± 0.002. This observation suggests an optimal length of the transmission line for maximum amplification, which is possibly a balance between two opposing mechanisms. First, a longer line

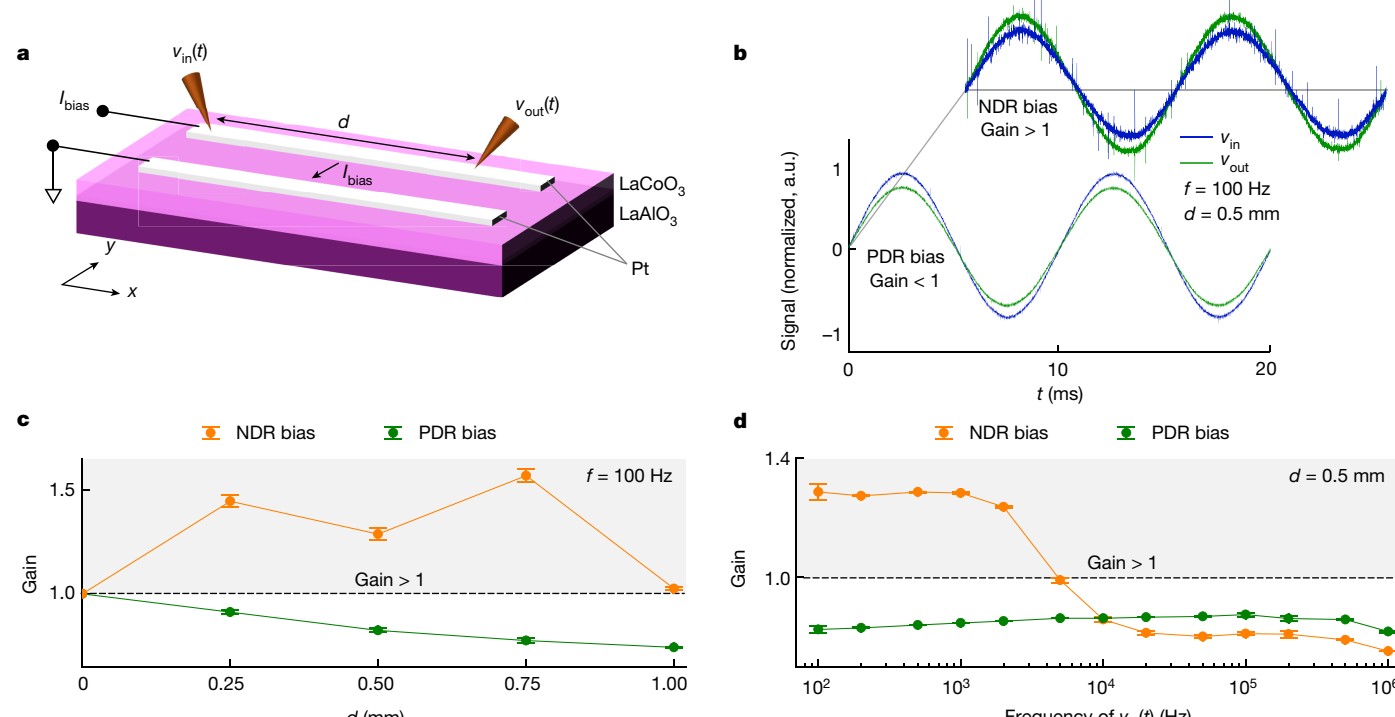

**Fig. 4 | Active transmission line. a**, Schematic of the active transmission line, and the applied and measured signals. The measurement displayed here was performed on a component with electrode length (in the $x$ direction) of 1 mm, with a gap between the electrodes (in the $y$ direction) of 12 μm. Output ($v_{out}(t)$) was measured at varying distances ($d$) from the input ($v_{in}(t)$) along the $x$ direction on the signal-carrying line at varying frequencies ($f$). **b**, Normalized time evolution of $v_{in}(t)$ and $v_{out}(t)$ on injecting either of two different $I_{bias}$ levels into the LaCoO$_3$ medium to poise it once in PDR (locally passive) and once in NDR (semi-stable EOC) regimes of the corresponding d.c. response of the structure ($f = 100$ Hz, $d = 0.5$ mm). For the NDR bias, it is apparent that $|v_{out}(t)| > |v_{in}(t)|$. Data have been normalized for ease of visualization such that $v_{in}(t)$ at different biases have the same normalized magnitude and the ratio between $v_{out}(t)$ and $v_{in}(t)$ at a given bias is maintained. **c**, Gain (amplitude of $v_{out}(t)$ divided by amplitude of $v_{in}(t)$) as a function of $d$ ($f = 100$ Hz). **d**, Gain as a function of $f$ ($d = 0.5$ mm). Error bars in **c** and **d** are smaller than most of the data markers. Estimation of error bars is described in the Methods.

allows a spatially larger amplification interface with the underlying semi-stable EOC medium, thereby promoting amplification. Second, a shorter line offers lower resistance for the amplification to overcome. As the frequency of the a.c. input signal increases, the amplification decreases and falls below unity gain at about 5 kHz (Fig. 4d shows an example dataset obtained at 0.5 mm from the input; the Methods provides additional details). In PDR bias, however, the gain was lower than 1 under any condition, and the signal amplitude decreased monotonically and exponentially with distance, with an attenuation of 2.0 ± 0.1 dB mm$^{-1}$ (Supplementary Fig. 16). In all our measurements on varying lengths of the transmission line (up to 4 mm), NDR bias in super-quadrature bandwidths always produced a larger gain compared with PDR bias, consistent with the observations presented here (Supplementary Figs. 14–16). Furthermore, to eliminate potential artefacts in the measurement setup or in the transmission-line structures, we repeated the measurements using nominally identical transmission lines assembled on SiO$_2$ (a passive medium), using a discrete resistor in place of the LaCoO$_3$ device and using discretized emulation composed of a distributed network of passive resistors with resistances similar to the transmission line. In all these cases, as expected, similar to the transmission lines operated in PDR, there was no over-unity gain under any conditions, and the signal amplitude decreased exponentially with distance from the input (Supplementary Figs. 17–20).

## Mechanism of energy conservation

Although signal amplification in the transmission line was consistent with our hypothesis, its physical validation requires a demonstration of its mechanism. Even though the theory establishes the connection between EOC biases and amplification of perturbations, the purely mathematical nature of the EOC theory makes it agnostic to the energy source of the amplification. Specifically, because there is more energy in the amplified a.c. signal than in the input a.c. signal, the additional a.c. power ($\Delta P_{a.c.}$) must come from the only available energy source—the d.c. power supplied to the device by the static current source.

To test this idea, we repeated the phase-shifting measurements in the test structures (Fig. 2a) under operando thermal mapping using a calibrated infrared camera (Fig. 5a). The thermal measurements were averaged over hundreds of cycles to extract sufficient signal levels. In the presence of a.c. signals, we expected super-quadrature phase shifts to accompany less net d.c. power dissipated into the device, leading to lower Joule heating and reduced average temperature. Therefore, we fixed the d.c. bias at semi-stable EOC, and varied the frequency of the input signal from 0 Hz (effectively no small signal) to 50 kHz. This covered a range of both sub-quadrature ($\Delta P_{a.c.} < 0$ or dissipation) and super-quadrature ($\Delta P_{a.c.} > 0$ or amplification) regions, and ensured that the total average input power was approximately constant. The measurement at 0 Hz ($\Delta P_{a.c.} = 0$) provided a baseline for the total d.c. power dissipated in the medium (Fig. 5a, 'd.c. only'). With an input signal with finite power at a frequency of 100 Hz, relative to the zero input signal, there was a lower temperature in the device, corresponding to reduced heat dissipation (or localized cooling).

A cross-section of the temperature map at 100 Hz, compared with the d.c.-only case, is displayed in Fig. 5b, and the temperature change obtained by such cross-sections is plotted across various measured frequencies (Fig. 5c). We repeated the measurement at a second (and

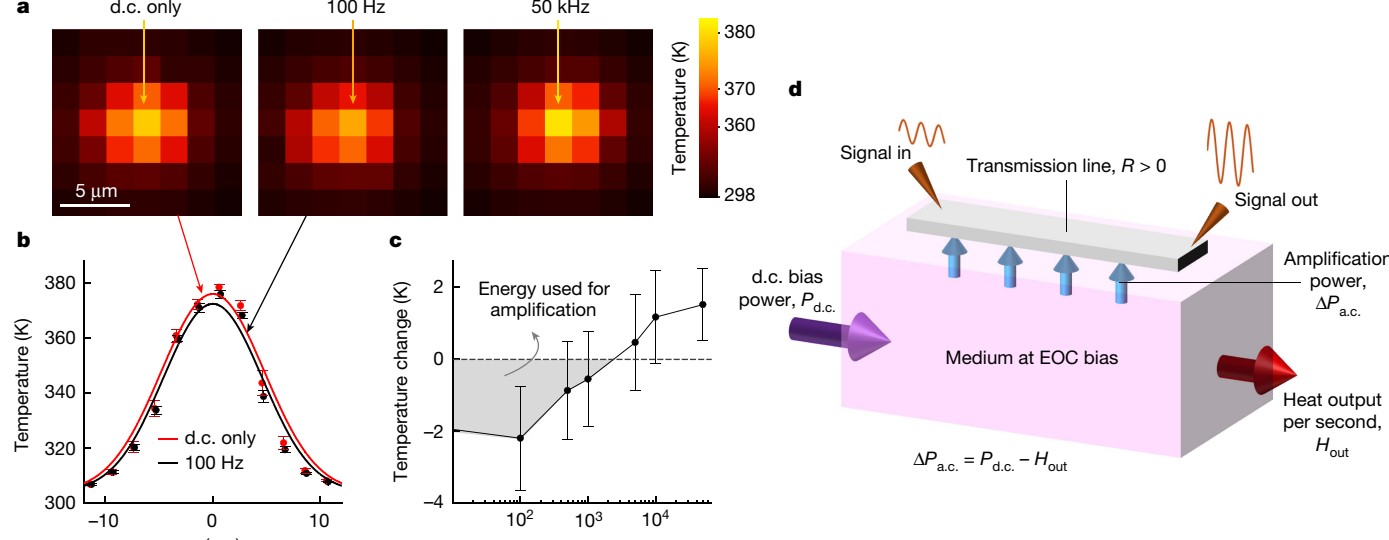

**Fig. 5 | Analysing energy partitioning in semi-stable EOC. a**, Black-body thermal maps of the active region of a test structure (illustrated in Fig. 2a) biased at semi-stable EOC ($I_{bias}$ = 3 mA), with three different temporal small signals superimposed on the same d.c. bias in semi-stable EOC. On careful review, it is apparent that the heat map corresponding to a signal of frequency 100 Hz exhibits lower temperatures than the other two (especially the centre pixel, which is indicated with an arrow). **b**, Cross-sections of the heat maps comparing the temperatures for d.c. bias with no signal (0 Hz) and with a 100 Hz input signal. Data markers correspond to experimental data and curves correspond to Gaussian fits. **c**, Change in local temperature corresponding to small signals of different frequencies relative to the local temperatures corresponding to d.c. bias, all of which are measured at the same d.c. bias in semi-stable EOC. The shaded region indicates lower temperatures or relative local cooling. The error bars in all the panels are described in the Methods. **d**, Mechanism of a part of the input d.c. energy being used for signal amplification when biased at semi-stable EOC.

higher) semi-stable EOC bias, with qualitatively similar results, but with a much higher bandwidth, indicating bandwidth tunability through bias (Methods). This observation is seemingly surprising because in all the passive electronic components, a larger input power leads to greater heat dissipation. However, in the case of semi-stable EOC, this observation is expected, and is direct evidence of our mechanistic interpretation of energy partitioning. That is, energy from the d.c. bias, instead of fully dissipating as heat (as happens in most electronic components), is used to amplify small signals in the transmission line (Fig. 5d).

## Discussion and observations

A central aspect of the experiments was our ability to isolate semi-stable EOC, or equivalently, non-oscillating NDR (by avoiding destabilization by parasitics). In our reformulation of EOC theory[7], we showed that a minimum degree of electrothermal nonlinearity is needed to induce NDR; however, the external reactance needed to destabilize and induce self-oscillations at EOC varies inversely as the strength of nonlinearity. The relatively gradual second-order change in conductivity in LaCoO$_3$ (by about three orders of magnitude) over approximately 300 K offered the minimum degree of nonlinearity required. By contrast, the nonlinearity in the change was also sufficiently low, which ensured that parasitics (typical circuit or device capacitances) would not destabilize the semi-stable EOC. Most neuronal devices built in the past used abrupt first-order phase transitions[31–33], which can be easily destabilized, leading to self-oscillations and obscuring of the effects shown here. Therefore, the use of the spin transition in LaCoO$_3$ was crucial.

Although EOC-like features are postulated during voltage-controlled NDR (the current decreases as the voltage is increased), for instance, in Esaki diodes, there is no experimental verification of semi-stable EOC nor an identification of the state variable(s)[44,45]. Similar to how the current-controlled NDR in LaCoO$_3$ is described by temperature as a state variable, to verify and exploit the potential semi-stable EOC

in voltage-controlled NDR systems, a dynamic state variable needs to be identified in such systems. Zero-resistance transmission resembles superconductivity, which usually requires specific ambient conditions (such as a temperature or a pressure). Active transmission, by contrast, offers controllability across positive, zero and negative resistances (but only to time-varying small-amplitude signals); it can be achieved under normal temperature and pressure; and it needs an electrically energized EOC medium.

Although our experimental system dissipated notable d.c. power (in the milliwatt range) on biasing in semi-stable EOC, it proved the principle of a previously undescribed and surprising primitive. Optimization for performance awaits a more detailed understanding of the d.c.–a.c. power conversion, the theoretical limits of local amplification, a means to increase the super-quadrature transmission frequency limit, and the identification of optimal materials and geometric configurations.

## Conclusion

Guided by an approach motivated by material properties to the local activity theory, we isolated semi-stable EOC in a spin crossover material, namely LaCoO$_3$, with weakly nonlinear thermally tuned electrical conductivity. Using a compact test structure, we observed the coexistence of a non-oscillating NDR region, super-quadrature phase shifts (between a.c. current and a.c. voltage) up to the theoretical maximum of π and the amplification of perturbations, which together confirmed the existence of the experimentally elusive semi-stable EOC. We constructed a continuous active transmission line coupled to a semi-stable EOC medium, where a signal propagating inside a resistive metallic line was amplified when the underlying medium was biased at semi-stable EOC. Finally, we showed, using operando thermal mapping, that the energy needed to induce amplification and active transmission comes from the d.c. power supplied to the device. This fundamentally different and surprising signal transmission primitive enables signal amplification in a spatially continuous and local manner, without breaking the signal-carrying conductor to perform amplification. Such a

solution, which potentially avoids thousands of repeaters and buffers, could greatly alleviate interconnect issues that bottleneck the current component-dense chips.

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

## Methods

### EOC nomenclature

EOC, by definition, refers to semi-stability. However, since previous reports have used the phrase 'edge of chaos' to describe destabilized manifestations of the underlying EOC, such as chaotic dynamics[43], here we refer to the distinct semi-stable behaviour exhibited by our devices as semi-stable EOC.

### Material growth and device fabrication

LaCoO$_3$ blanket films were grown on a LaAlO$_3$ substrate through pulsed laser deposition using a Nd:YAG (neodymium-doped yttrium aluminum garnet) laser to ablate a stoichiometric LaCoO$_3$ ceramic target, at a wavelength $l = 266$ nm, operated at 5 Hz. The oxygen partial pressure of the chamber was 100 mtorr and the substrate temperature was maintained at 650 °C during growth. The as-grown films were then annealed in air at 500 °C for 1 h to reduce oxygen deficiencies. Although Fig. 2b refers to the state of LaCoO$_3$ at 300 K as low spin, since the spin crossover in LaCoO$_3$ begins well below room temperature, the state at 300 K−relative to that at 600 K−is a lower-spin state.

An MA6 mask aligner and an MLA150 maskless aligner were used to fabricate the patterned electrodes on blanket LaCoO$_3$ films. The AZ5214E reversible photoresist was used in its image reversal process. After baking at 110 °C, the resist received 50 mJ cm$^{-2}$ exposure and, after exposure was baked at 115 °C. Then, it received a flood exposure of 900 mJ cm$^{-2}$ and was developed in AZ400K-1:4 for 1 min. Then, 100 nm of Pt electrodes (with an adhesive Cr or Ti layer of less than 5 nm) were sputter deposited followed by lift-off in Remover PG at room temperature assisted by mechanical agitation. The electrodes were also deposited by e-beam evaporation in some of the samples, with similar results. The electrode gap (8–12 µm) in the specific structure shown in Fig. 4a and the metal thickness were chosen for the ease of fabrication using optical lithography, and the lengths of the transmission line (varying from 0.2 to 4 mm) were chosen to represent typical chip-layout dimensions.

### Electrical measurements

**Setup.** The devices were tested using FormFactor ACP40-GSG-250 radio-frequency probes connected to the d.c. + a.c. source current through subminiature version A connectors and mini coaxial cables totalling a length of less than 1.2 m. Electrical phase-shift measurements were directly performed using an a.c. voltage supplied and measured by an Agilent 33220A 20 MHz function generator and an Agilent InfiniiVision MSO7054A 500 MHz oscilloscope, with d.c. power supplied by a Keysight B2911A source measure unit (SMU), with the spot measure time manually set to greater than 300 ms (to fully approach the steady state) and auto-ranging turned off (to avoid range-switching artefacts).

**Repeatability of I–V behaviour.** Quasistatic current-controlled *I–V* sweeps were repeated for a variety of device dimensions under identical conditions, using identical replicates, and with more than six consecutive *I–V* sweep cycles (Supplementary Fig. 5). Although there are quantitative differences from the first cycle to the second and subsequent cycles, all of them are qualitatively similar. There was typically negligible quantitative variation between cycles after the first cycle. For example, the range in voltage at 2 mA varies by 20 mV (0.1%) for the 3 µm device (Supplementary Fig. 5a), and by 260 mV (0.7%) for the duplicate 10 µm device (Supplementary Fig. 5f). The consistency and repeatability of these measurements verifies that the observed NDR was robust and probably not due to artefacts such as noise or gradual device damage. The NDR-onset current bias was identified using Savitzky–Golay smoothing and differentiation (Supplementary Fig. 6). For transmission-line structures, the cycle-to-cycle variation was higher, probably due to the inhomogeneous nucleation of phase transition throughout the much larger active volume.

### Power supply and test circuit

**Setup.** Because LaCoO$_3$ devices manifest current-controlled NDR, measuring frequency-dependent phase shifts as a function of current bias required coupling tunable d.c. current and a.c. current sources. It was also necessary for the two to be reactively decoupled so that the reactance of the d.c. source would be screened from the a.c. source. We achieved this setup by coupling a Keysight SMU in the current-source mode (the same one used for *I–V* sweeps) to an Agilent function generator using a diode and 200 and 5.1 kΩ resistors (Supplementary Fig. 7a). We found that the diode effectively screened the SMU without introducing artefacts so long as the diode was fully forward biased, for currents greater than 0.1 mA.

The 200 kΩ resistor simultaneously performed two crucial roles. First, it converted the signal from the function generator from 5 V to less than 25 µA, so that the signal was small with respect to the d.c. bias. Second, it generated a sufficient non-reactive load such that the voltage from the function generator and the current through the circuit and device were in phase with an error of 0.01 radians, so that the function generator voltage is a valid substitute for the circuit current with respect to computing phase shifts. This ability to take the supplied a.c. voltage as essentially in phase with the a.c. current through the circuit as well as the LaCoO$_3$ was experimentally necessary, because the current through the circuit was too small and noisy to reliably trigger the oscilloscope and varied compared with the d.c. current. By contrast, the function generator always produced a reliable noise-free 5 V (a.c.) signal. The 5.1 kΩ resistor was used to dampen oscillations during *I–V* sweeps and was retained during subsequent measurements for consistency. This test circuit was used for all the measurements of the two-terminal test structures, including during operando thermal mapping.

For the transmission-line measurements, larger voltages were required; therefore, the 200 kΩ resistor had to be removed. Without the resistor, the voltage from the function generator was too large to fully forward bias the diode. Therefore, the diode was replaced with a capacitor to provide a d.c. block between the function generator and the d.c. source (Supplementary Fig. 7b). The 5.1 kΩ resistor was similarly replaced with a 1.1 kΩ resistor to reduce the total d.c. voltage sourced from the SMU from around 50 to about 30 V (as the transmission lines are more conductive structures and therefore operate at higher d.c. currents).

**A common measurement pitfall.** Reactive phase shifts are routinely measured using black-box impedance analysers. However, we measured the dynamic properties of our nonlinear devices by directly analysing their response to a.c. signals from a function generator recorded on an oscilloscope, which we have found to be more reliable for three reasons. First, we could directly observe that the dynamic signals were small enough to minimize distortion and power lost to higher harmonics. Second, a direct signal-versus-time measurement can be modelled with standard physics-based simulators. Third, we could account for all the parasitic reactive components in the measurement system in our models. A black-box impedance analyser used for such measurements often leads to highly erroneous results. For example, a measurement of $\Delta\phi \approx \frac{\pi}{2}$ at 100 Hz when biased at the cusp of NDR (Fig. 3c; raw data are shown in Supplementary Fig. 8) would be interpreted as a pure inductor with an unreasonable magnitude of several millihenry. Similarly, $\frac{\pi}{2} < |\Delta\phi| \leq \pi$ at 100 Hz for bias currents of 3–4 mA would be interpreted as the presence of an actively powered circuit like an operational amplifier. These anomalies also confounded Hodgkin and Huxley in their initial investigations of the neuron[16]. Thus, it is important to carefully understand and analyse phase shifts in nonlinear components.

### Signal processing

**Sinusoidal amplitudes and phases.** Amplitudes $A$ and absolute phases $\phi$ (with respect to cosine) of the sinusoidal signals were manually

computed from raw oscilloscope time series $s(t)$ by taking their inner product with a cosine and sine at the base frequency $f$, and normalizing by the number of complete cycles $N$ as

$$A = \sqrt{C^2 + S^2}; \tan\phi = -\frac{S}{C}, \tag{1}$$

$$C = \frac{2f}{N} \int_0^{\frac{N}{f}} s(t) \times \cos(2\pi ft)\,dt, \tag{2}$$

$$S = \frac{2f}{N} \int_0^{\frac{N}{f}} s(t) \times \sin(2\pi ft)\,dt. \tag{3}$$

To illustrate the uncertainties in the phase contour plot (Fig. 3c), Supplementary Fig. 9 displays the cross-sections of Fig. 3c.

**Electrical uncertainty quantification.** For sinusoidal phases and amplitude, uncertainty was estimated with leave-one-out cross-validation. That is, temporarily removing one of the $N$ cycles ($N > 15$) from the dataset, repeating the amplitude–phase calculation for the remaining $N - 1$ cycles and then taking the standard deviation of the distribution. Uncertainties in gain ratios and phase differences were propagated from amplitudes and phases in the standard way. Typical phase uncertainties were found to be of the order of 0.01 radians—too small to be visible in most plots. Similarly, uncertainties in the gain were also too small to be visible in most plots.

**PSDs.** PSDs were computed using the fast Fourier transform implementation of MATLAB R2021B, and a Hann windowing function was used to eliminate spectral leakage before normalizing by the Nyquist frequency. Welch averaging with three sections and 30% overlap was used to trade some spectral resolution of the periodograms for improved statistical variation in the fast Fourier transform amplitudes. The frequency resolution after Welch averaging was better than 1.5 Hz.

The data in Fig. 3d were normalized by the 100 Hz frequency component to decouple the amplification from the expected Johnson–Nyquist scaling. The PSD of the Johnson–Nyquist noise has a known scaling with temperature $T$ and local resistance $dR$ as

$$PSD_{JN} = k_B T\, dR. \tag{4}$$

Here $k_B$ is the Boltzmann constant. Thus, normalizing the periodograms by the PSD at one fixed frequency component eliminates the Johnson–Nyquist scaling, as for a periodogram at a given bias, all the components have the same $T$ and $dR$ values. When comparing the data before and after normalization (Supplementary Fig. 10), the most important features of the data are preserved. In particular, the local maxima in the noise at 60 Hz and their harmonics for current biases near the NDR onset at roughly 3 mA are in both raw and normalized data and therefore not an artefact of normalization. By contrast, the raw data exhibit a rapid decrease in absolute magnitude as the bias approaches 3 mA from either direction. This is a direct effect of the expected Johnson–Nyquist scaling, since the differential resistance also rapidly decreases near the NDR onset. The rapid decrease in the noise baseline obscures the shape of the amplification peak. Thus, normalizing by the PSD at 100 Hz, as we have done in the main text, clarifies the features of the amplification peak without introducing artefacts.

The cross-sections of the PSDs (Supplementary Fig. 11a,b) at a fixed bias current display the detailed features of the periodograms. The most obvious feature is the largest peak at 100 Hz corresponding to the input sinusoid, but the data also clearly show a strong peak at 60 Hz and its harmonics (along with a peak at 1 kHz), which is only present in the data after the NDR onset at roughly 3 mA. The cross-sections

at a constant frequency (Supplementary Fig. 11c) show that the noise amplification of about 20 dB exists over a large bandwidth.

**Transmission-line measurements**

**Shape of the $I$–$V$ curve.** The quasistatic $I$–$V$ curves corresponding to both test structures and transmission lines (Supplementary Figs. 5, 12, 14 and 15) exhibit an abrupt jump around (either before or after) the onset of NDR. This behaviour is probably associated with the abrupt localization of spatial thermal gradients, driven by energy minimization principles, which was first theorized in 1936 (ref. 46) and formalized later[47]. Although this effect deserves a dedicated study, owing to the increasing nonlinearities in current nanoscale electronic components, in our experiments, we verified that the abrupt jump had no effect on our observations. Specifically, our measurements show that EOC and amplification are associated with the onset of NDR and not the abrupt features in the $I$–$V$ curves. The transmission lines had an end-to-end resistance of 6.6 kΩ mm$^{-1}$ (measured at 0.1 V) and the 1 mm line had a signal-to-ground resistance of 1 kΩ.

**Insertion delays.** To eliminate potential insertion delays, we characterized the delays induced by the setup by placing two probes on the same pad of a transmission line, which produced delays smaller than any delays measured across different pads on the same transmission line (delays were less than $0.003 \times 2\pi$ radians).

**Thermal-mapping measurements**

**Setup.** Operando thermal maps were acquired with an FLIR SC6700 infrared camera and a 120 Hz acquisition rate. Raw data as the infrared signal were calibrated to temperature (Supplementary Fig. 21) using independent isothermal data obtained with an Instec HCP421V-MP+ temperature stage, which was independently verified to 1 K accuracy with a Pt metallic line thermometer. We note that the calibration data extended to 400 K, greater than any of the experimental data (measured to a maximum of 370 K); the calibration provided only interpolation, not extrapolation. Composite thermal images were obtained as pixel-wise averages over >30 frames from the thermal video. Gaussian fits to thermal profiles were computed by a least-squares analysis.

**Reproducibility of the thermal measurements.** The thermal-mapping experiment was attempted at several PDR biases, where—in contrast to the results shown in Fig. 5—only net heating would be expected. However, the observed temperature change was not statistically different from zero for the different cases (with and without a small signal in addition to the bias). This issue is an instrumental limitation: because the baseline temperature in the PDR is substantially lower, the signal to the IR camera is exponentially smaller.

Net device cooling during super-quadrature phase shifts (Fig. 5) is a substantial claim. Hence, we verified it by duplicating the result at a second independent NDR bias ($I_{bias} = 4.1$ mA) (Supplementary Fig. 22). In contrast to Fig. 5 (at $I_{bias} = 3$ mA, where the maximum frequency at which relative cooling could be observed was about 2 kHz), for $I_{bias} = 4.1$ mA, this frequency was more than 10 kHz. This observation indicates a notable increase in the bandwidth on slightly changing the bias.

**Thermal uncertainty quantification.** Uncertainties in thermal data were estimated as the standard deviation of more than 30 frames from the FLIR thermal video and then propagated using the temperature-calibration power laws. Uncertainties in the Gaussian-fit peak temperatures were estimated by bootstrapping. For each measured point $(x, T)$ with measured uncertainty $\delta T$ in the cross-section, a normal distribution $\mathcal{N}(T, \delta T)$ was generated, a sample was selected at random from the distribution and finally the new sampled data were fit with Gaussians, and the Gaussian fits were recorded. The sampling process was repeated for $N = 5{,}000$ times, and the uncertainty in the peak of

the Gaussian fit was taken as the standard deviation of the 5,000 bootstrap simulations.

## Data availability

The data supporting the findings of this study are available in the Article and its Supplementary Information. Source data are provided with this paper. To promote rapid dissemination and reproduction of this research, a limited number of devices used in this study have been saved and will be supplied upon request to the corresponding author for scientific studies, as long as their results will be published.

## Code availability

The codes used for modelling in this study are available as Supplementary Data 1 along with basic information on running the codes.

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

**Acknowledgements** We thank G. Bhatt, R. Midya and E. Salagre for help with fabricating the SiO$_2$ transmission lines, and S. Goyal for assistance with the data collection and proofreading the manuscript. This work was primarily supported as part of the Center for Reconfigurable Electronic Materials Inspired by Nonlinear Neuron Dynamics (reMIND), an Energy Frontier Research Center (EFRC) funded by the US Department of Energy (DOE), Office of Science, Basic Energy Sciences. The Laboratory Directed R&D (LDRD) programme of Sandia National Laboratories provided internal support to the reMIND EFRC. All authors of this work were supported directly by reMIND or by the LDRD programme. E.P. was partially supported by the SystemX Alliance at Stanford. Part of this work was performed at the Stanford Nano Shared Facilities (SNSF), supported by the National Science Foundation under award ECCS-2026822. Sandia National Laboratories is operated for the US DOE's National Nuclear Security Administration under contract DE-NA0003525. This paper describes objective technical results and analyses. Any subjective views or opinions that might be expressed in the paper do not necessarily represent the views of the US DOE or the United States Government.

**Author contributions** R.S.W. conceived the idea of an axon-like active transmission line in 2012, and envisioned biasing near an electronic phase transition to realize it. R.S.W. and S.K. investigated various material candidates, and identified the material properties required to isolate the EOC. T.D.B., R.S.W. and S.K. designed the experiments and wrote the initial draft of the manuscript. All authors approved the final version of the manuscript. T.D.B. performed all the electrical measurements, supported by F.U.N. and E.D.G. A.Z. and E.J.F. grew the films and performed the material characterization. J.L.C. and P.J.S. constructed the compact model and performed the simulations, with guidance from T.D.B. T.D.B. performed the thermal measurements, with advice from A.A.T. and support from E.D.G., J.Z. and S.R. M.I. fabricated the electrodes on the oxide films, with advice from E.P. S.K. oversaw the project.

**Competing interests** The authors declare no competing interests.

**Additional information**
**Correspondence and requests for materials** should be addressed to Suhas Kumar.
