## [Peer Review file · Nature]

Manuscript Title: Axon-like Active Signal Transmission

Reviewer Comments & Author Rebuttals

Reviewer Reports on the Initial Version:

Referee #1 (Remarks to the Author):

The manuscript by Brown and coworkers describes an experimental study showing signal transmission without losses using a media with NDR, reminiscent of active media. The work is highly exciting and, if correct, would clearly motivate further studies, both in fundamental science and technology. To my knowledge these are rather new and untested ideas. I very much hope the authors can address my (and other reviewers' comments) and see this work published.

The main result is a demonstration of signal amplification in the NDR regime of a transmission line geometry. The methods as such are suitable, the authors appear to have made proper experiments and sample fabrication according to scientific standards. The thermal measurements are trustworthy although described painfully short. The circuitry is described in sufficient detail to allow reproducing the measurements.

However, I find the work rather difficult to read and not easy to follow. Also some basic reasoning and questions appear to be omitted (or I failed to find them the documents), see below. In the following I will list several comments and suggestions for changes that might improve clarity.

1) The main question I have relates to the mechanism proposed and supported by the modelling. Namely, there is a compact model with lumped elements considered, while the experiment is fundamentally a type of a lossy transmission-line. Accordingly, I was expecting a characterization of the resistance (total and per unit length) across LCO film between the electrodes (arrangement in fig 4a)

2) A signal transport measurement along a transmission line is not very trivial. I am sure the authors are aware of the issues of frequency dependent reflection at the ends of the line etc, frequency response of signal insertion and extraction, etc.. This should be addressed. I find a characterization of distance dependent signal amplitude essential, which should show a damping e.g. exponentially (depending on parameters).

To be certain to not have measured artefacts, a reference measurement using the same electrode geometry by a trivial SiO₂ substrate is strongly recommended.

3) Overall the description of the sample should give more detail or should be more precise. For example it is not clear for me why the dimensions of device in fig 4a was chosen.

4) The discussion around figure 3 is a little confusing. There is talk about noise amplification in the NDR region. In the 3mA trace in fig3b there is clearly noise, which appears by eye have a strong frequency component around 1kHz. However, the plot 3d cuts at exactly 1 kHz (while 3c shows up to 10kHz). Why was it cut? What is the noise in 3b?

5) On page 7 there is reference to figure 2d, which does not exist.

6) The experiment shown in figure 4 sets the voltage to be in the NDR range, in contrast to setting the current in fig3. This makes me wonder: what is the current to the other electrode, what is the

meaning of NDR in voltage-controlled setting (see also comment below)? The reader is left alone with such questions.

7) What is the thermal time constant of the device (relating to the thermal measurements of fig 5 and SI)?

I am further listing some minor points. These are for the consideration of the authors but without mandatory change expected.

- Page 4: In principle the NDR can be hosted in both S-shaped and N-shaped IVs, and then accessed via current or voltage source. Here only current-driven system are considered. Would the idea hold for a voltage driven system?

- Page 3, references 26-31 appear unnecessary self-citing, given that many other groups have contributed to the field. This does not help my confidence that the authors have not missed anything concerning this current work.

- I did not understand why the authors prefer to work in one measurement range, to avoid stitching effects. If they exist they question the accuracy of any given range. It appears a pity why noisy low frequency data is shown when there could be a more accurate low frequency separate measurement set.

- I would be interested to see a line trace similar to fig 4b for the point at 4-5kHz.

Referee #2 (Remarks to the Author):

The paper describes the experimental verification of active signal amplification. The authors use LaCoO₃ as the host material for realizing the edge of chaos (EOS) state, especially the semi-stable EOS. Formerly, they used VO₂ as the host material for the EOS, where the first-order insulator-metal transition triggers a more unstable reaction. This time they used LaCoO₃ because it shows a more gradual change of resistivity as a function of temperature and as well as a function of applied current. They verify the signal amplifications in 3 steps, namely, negative differential resistance (NDR), phase shift, and noise amplification. They further tested the active amplification using a real device. They successfully observed the amplified signal in the NDR regime, whereas they observed signal dissipation in the positive differential resistance (PDR) regime. The gain of the signal slightly exceeds unity. They further verify using a thermal imaging technique that a part of the energy input is used for the amplification of the signal of the transmission line. The paper addresses the application of the strongly correlated electron material to the electronic device mimicking the signal amplification in biological systems, which is very interesting.

From the material science point of view, I tried to understand the origin of the negative differential resistance in LaCoO₃. Fig. 2c shows data only up to 4 mA of I_{bias}, which makes me wonder what is happening I_{bias} > 4 mA. I looked at Fig. S3 in the supplemental material, which shows even more puzzling data where we see an abrupt change in the I-V curve. In VO₂, similar data is obtained which are attributed to a Mott transition. In LaCoO₃, people found no feature of sudden transition as a function of temperature. Therefore, the sudden change in the I-V curve is interesting and puzzling to me.

I believe that material science is not the first priority of the paper, but the proof of principle for the active signal amplification is. However, their data is already interesting from a condensed matter physics point of view. I hope the authors address this issue before the publication of the paper.

The above point weakens their conclusion to me. Except for that, the paper is clearly written. The figures are also clear. The result and the logic are interesting. I thus recommend its publication only after they adequately addresses the issue.

Referee #3 (Remarks to the Author):

Active Signal Transmission at the Edge of Chaos

A. Summary of the key results

As a major novelty in science, the authors demonstrate how a small signal may be amplified over a certain frequency range as it travels across a purely-passive transmission line, if the latter is integrated on top of a substrate, composed of a new material, specifically LaCoO₃, biased on the negative differential resistance (NDR) region of its DC voltage (V) versus current (I) characteristic.

B. Originality and Significance

The paper reports a novel, original and highly-significant research work. It deserves publication on Nature in its current form.

Interestingly, a two-terminal memristor, composed of the new material, may be stabilized easily along the NDR region of its V-I locus, since, due to its mild conductivity versus temperature nonlinearity, is insensitive to the reactances in the DC measurement set-up. Moreover, when biased along the NDR region, its small-signal voltage and current signals appear to admit a relative phase difference ranging from $\pi/2$ to π in modulus. Using such a material to form an excitable substrate, lying under a purely-dissipative transmission line, the small signals respectively at the output and at the input of the transmission line itself were found to be shifted one relative to the other by at least $\pi/2$ and by at most π in modulus, while the output signal was found to be larger than the input signal across a certain frequency range, when the substrate was continuously polarized on the Edge of Chaos.

Very importantly, employing operando thermal mappings, it was further demonstrated how the energy, necessary for the amplification of the small signal across the transmission line, comes from the DC power supplied to the LaCoO₃ medium.

This remarkable research work may pave the way toward the future insertion of excitable media, biased on the Edge of Chaos, below lengthy on-chip transmission lines, to induce small signal amplification across them without the use of repeaters and buffers.

C. Data & methodology: validity of approach, quality of data, quality of presentation

This paper is an example of excellence in research.

It provides a comprehensive justification to the breakthrough research findings in the lab through a deep theoretical analysis corroborated by insightful model simulations.

It deserves to be published in Nature without further revision.

The authors of this manuscript include some of the most reliable and trustworthy device scientists on earth, and this adds credibility and solidity to the theoretico-experimental work.

Moreover, the paper is well written and the discussion of the research work is crystal clear.

D. Appropriate use of statistics and treatment of uncertainties

Not applicable

E. Conclusions: robustness, validity, reliability

The authors have conducted a large number of additional experiments, besides those shown in the main manuscript, as reported in the supplementary information file, confirming the robustness of the phenomenal research findings.

F. Suggested improvements: experiments, data for possible revision

The following list of suggestions are very minor comments, which do not need another round of revision:

As a suggestion, it would be nice to add a short section, or even a brief remark, formalizing the difference between Edge of Chaos and semi-stable Edge of Chaos, already within the manuscript.

Caption of Fig. 1: negative differential behavior-> negative differential resistance behavior

Page 3 steady changes -> gentle changes

Last line at page 5: (Fig. 3b) -> (Figs. 3b and 3c)

Fig. 3(c): replace π with $-\pi$ on the bottom of the color map

Fig. 3(c): add the indication of I_{bias} along the x axis, as done in Fig. 3(d)

Caption of Fig. 3(c): is displayed in Fig. 2d -> is displayed in Fig. 2c

In regard to the measurements on the circuit of Fig 3(a), it was said that, differently from what was done in the experiments on the test structure of Fig. 2(a), the voltage V_{bias} (v) served as the bias (small-signal) input variable, while the DC (small-signal) current I_{bias} (i) was employed as the output variable. Use one sentence to confirm whether the small-signal current i in the circuit of Fig. 3(a) was then converted back to some voltage v_{out} . This curiosity arises from the inspection of Fig. 4(a), referring however to another circuit set-up, where the output signal, specifically v_{out} , appears in voltage form.

all three criteria three criteria-> all the three criteria

Page 8. Where it says "amplification decreased and fell below unity gain at about 4 kHz (Fig. 4c)" should 4 kHz rather be 10 kHz?

Page 8. Where it says "as observed here (roughly 70 kHz)", should it maybe be "as observed here (roughly 50 kHz)", as mentioned some lines above?

Caption of Fig. 4: “Normalized time evolution of input ($v_{in}(t)$) and output ($v_{out}(t)$) for two different I_{bias} – one in PDR (locally passive) and another in NDR (semi-stable EoC) of the corresponding DC response of the structure” -> Normalized time evolution of the input ($v_{in}(t)$) and output ($v_{out}(t)$) signals of the transmission line upon injecting either of two different currents (I_{bias}) into the LaCoO₃ substrate so as to poise it once in the PDR (locally passive) regime and once in the NDR (semi-stable EoC) regime.

Page 12. It is said that the temporal data in Fig. 4b include the transmission line AC current responses to small signal stimuli under bias currents of 3mA and 4mA. First of all, the AC responses are shown in voltage form and not in current form, right? Secondly, did you really show the transmission line AC responses for both the bias current levels 3 mA and 4mA in Fig. 4b? Please check it.

G. References: appropriate credit to previous work?

References to some other recent work on Edge of Chaos from the literature – e.g. the research studies on symmetry-breaking phenomena in homogeneous cellular media as well as on the impact of the Sodium (Na) ion channel on the emergence of the All-to-None complex neuronal phenomenon - could make the bibliography more comprehensive.

H. Clarity and context: lucidity of abstract/summary, appropriateness of abstract, introduction and conclusions

The paper is very clearly written. I believe the text and presentation of the work, including the illustrations, could not be done any better.

Author Rebuttals to Initial Comments:

Your suggestions have truly improved the content of the manuscript. We are pleased with the overall positive reviews. Nonetheless, we have meticulously addressed your suggestions by performing additional work and detailed edits, as necessary.

Owing to the additional experiments, some of our observations are now much clearer. For example, we now report gain up to 1.7 with negligible error bars (in our initial observation, it was up to 1.055). As such, we now need fewer analyses to substantiate our observations. Partly because of this simplification and because we moved parts of the main text to the methods section, our main text is less than 3500 words in length. To comply with formatting guidelines, we have moved the figures to the end of the main text.

We have provided a point-by-point response to all your comments in the following pages, which we believe you will find sufficient. If you think the paper can be improved further, we will be happy to hear additional comments.

Referees' comments:

Referee #1 (Remarks to the Author):

The manuscript by Brown and coworkers describes an experimental study showing signal transmission without losses using a media with NDR, reminiscent of active media. The work is highly exciting and, if correct, would clearly motivate further studies, both in fundamental science and technology. To my knowledge these are rather new and untested ideas. I very much hope the authors can address my (and other reviewers' comments) and see this work published.

The main result is a demonstration of signal amplification in the NDR regime of a transmission line geometry. The methods as such are suitable, the authors appear to have made proper experiments and sample fabrication according to scientific standards. The thermal measurements are trustworthy although described painfully short. The circuitry is described in sufficient detail to allow reproducing the measurements.

We have now written a detailed methods section (following the main text), which provides sufficient details on the thermal measurements.

However, I find the work rather difficult to read and not easy to follow. Also some basic reasoning and questions appear to be omitted (or I failed to find them the documents), see below. In the following I will list several comments and suggestions for changes that might improve clarity.

1) The main question I have relates to the mechanism proposed and supported by the modelling. Namely, there is a compact model with lumped elements considered, while the experiment is fundamentally a type of a lossy transmission-line. Accordingly, I was expecting a characterization of the resistance (total and per unit length) across LCO film between the electrodes (arrangement in fig 4a)

2) A signal transport measurement along a transmission line is not very trivial. I am sure the authors are aware of the issues of frequency dependent reflection at the ends of the line etc, frequency response of signal insertion and extraction, etc. This should be addressed. I find a characterization of distance dependent signal amplitude essential, which should show a damping e.g. exponentially (depending on parameters).

To address both comments (1) and (2), we fabricated new transmission line structures with varying lengths (from 0.25 mm to 4 mm). Further, each transmission line structure was designed with

intermediate electrodes, which allowed us to characterize the resistance at different distances from the input electrode (Supplementary Fig. 12, optical micrograph reproduced below).

Response Fig. 1. Newly fabricated transmission line structure, which enables probing of the signal at different points along the transmission line (included as a part of Supplementary Fig. 12).

For each length, we characterized the current-voltage behavior. Further, for each length, we also measured the transmission line behavior for NDR and PDR biases at varying intermediate distances (from the input electrode), and at varying frequencies.

We verified that: (1) In all cases, NDR bias produced a higher gain compared to PDR bias. (2) The signal amplitude always decreased with length in PDR bias (confirming the reviewer's prediction). (3) Interestingly, for NDR biases, there seems to be an optimal distance from the input where the gain is maximized (after which it drops at farther distances, as predicted by the reviewer).

Further, we have characterized the signal insertion delay (by placing two probes on the same pad), which produced delays smaller than any delays measured across different pads on the transmission line.

Changes: We have included a discussion on all these aspects in the main text, along with details in the Methods and the Supplementary Material.

To be certain to not have measured artefacts, a reference measurement using the same electrode geometry by a trivial SiO₂ substrate is strongly recommended.

We agree that artefacts are easy to introduce within transmission lines, and reference measurements are necessary. As such, we fabricated two sets of reference transmission lines: (1) Identical structures on a passive substrate (SiO₂), as suggested by the reviewer. (2) A distributed network of discrete passive resistors. In both cases, under measurement conditions similar to those used in the active transmission experiments, we observed no unexpected behaviors (or gain).

Changes: We have included a discussion on all these aspects in the main text, along with details in the Supplementary Material.

3) Overall the description of the sample should give more detail or should be more precise. For example it is not clear for me why the dimensions of device in fig 4a was chosen.

We have provided a detailed sub-section in the new Methods section, which offers details on how the devices were fabricated, how the test circuits were constructed, etc. Further, since we have repeated the measurements across different lengths of the transmission line (thanks to the reviewer's comment), we are not anymore choosing a specific length. Instead, we have made a

comment in the main text that this range of lengths was chosen to represent typical chip layout dimensions.

Changes: a detailed methods section and comments in the main text.

4) The discussion around figure 3 is a little confusing. There is talk about noise amplification in the NDR region. In the 3mA trace in fig3b there is clearly noise, which appears by eye have a strong frequency component around 1kHz. However, the plot 3d cuts at exactly 1 kHz (while 3c shows up to 10kHz). Why was it cut? What is the noise in 3b?

We realize this must have been confusing. The frequencies displayed in Fig. 3c and 3d refer to two different quantities (3c referring to the input voltage frequency, while 3d referring to the Fourier component in the measured current). We have now displayed Fig. 3d up to 10 KHz. To provide a better illustration of the different details, we have provided cross sections of the contour plot in Supplementary Fig. 11. The cross sections offer a detailed view into the various frequencies that are amplified. While there are some specific frequencies (such as 60 Hz and 1 kHz) that are amplified in NDR biases, there is an overall increase in the power spectral density at all frequencies at biases following the onset of NDR.

Changes: We have rescaled Fig. 3d. We have provided a discussion of these aspects in the Methods section, along with Supplementary Fig. 11.

5) On page 7 there is reference to figure 2d, which does not exist.

We have made this correction (changed to Fig. 2c). Thank you for catching the error.

6) The experiment shown in figure 4 sets the voltage to be in the NDR range, in contrast to setting the current in fig3. This makes me wonder: what is the current to the other electrode, what is the meaning of NDR in voltage-controlled setting (see also comment below)? The reader is left alone with such questions.

Thank you for catching a potentially confusing illustration. We always set the bias using a current. However, we also marked a bias voltage on the illustrations, since a bias current also naturally results in a specific bias voltage. We realize that this marking could be confusing, so we have now displayed only a bias current, and clarified in the text as well.

Changes: We fixed the illustration in Fig. 4 and clarified the text.

7) What is the thermal time constant of the device (relating to the thermal measurements of fig 5 and SI)?

We have experimentally calibrated our model, which provides both the thermal capacitance and the thermal resistance (in the form of thermal conductance in Supplementary Fig. 2). Using these quantities, we estimated the thermal time constant to be 0.6 ms to 5 ms. We have mentioned this in the section on Modeling in the Supplement.

Changes: We have provided the thermal time constant in Supplementary section 2.

I am further listing some **minor points**. These are for the consideration of the authors but without mandatory change expected.

- Page 4: In principle the NDR can be hosted in both S-shaped and N-shaped IVs, and then accessed via current or voltage source. Here only current-driven system are considered. Would the idea hold for a voltage driven system?

We think similar results are possible in N-NDR systems. In the main text's discussion section, we have written a brief paragraph on that, and outlined what would be required to achieve active transmission in an N-NDR system (i.e., identification and design of the correct state variable).

- Page 3, references 26-31 appear unnecessary self-citing, given that many other groups have contributed to the field. This does not help my confidence that the authors have not missed anything concerning this current work.

This is a fair criticism. We have replaced all of these references (except for our review paper) with references of papers produced by different groups around the world.

- I did not understand why the authors prefer to work in one measurement range, to avoid stitching effects. If they exist they question the accuracy of any given range. It appears a pity why noisy low frequency data is shown when there could be a more accurate low frequency separate measurement set.

We clarify that we are not performing any stitching of data. The reason to use the low amplitude is to avoid issues such as introduction of harmonics, reflections/ringing, disruption of the DC bias, etc., all of which would complicate the analysis of a transmission line (consistent with the reviewer's Comment (2)). We have mentioned these issues in the Methods section.

- I would be interested to see a line trace similar to fig 4b for the point at 4-5kHz.

We have added this data to Supplementary Fig. 13.

Referee #2 (Remarks to the Author):

The paper describes the experimental verification of active signal amplification. The authors use LaCoO₃ as the host material for realizing the edge of chaos (EOS) state, especially the semi-stable EOS. Formerly, they used VO₂ as the host material for the EOS, where the first-order insulator-metal transition triggers a more unstable reaction. This time they used LaCoO₃ because it shows a more gradual change of resistivity as a function of temperature and as well as a function of applied current. They verify the signal amplifications in 3 steps, namely, negative differential resistance (NDR), phase shift, and noise amplification. They further tested the active amplification using a real device. They successfully observed the amplified signal in the NDR regime, whereas they observed signal dissipation in the positive differential resistance (PDR) regime. The gain of the signal slightly exceeds unity. They further verify using a thermal imaging technique that a part of the energy input is used for the amplification of the signal of the transmission line. The paper addresses the application of the strongly correlated electron material to the electronic device mimicking the signal amplification in biological systems, which is very interesting.

Thank you for your comments. Following additional experiments (as suggested by Reviewer 1), unlike our previous data that exhibited a gain that “slightly exceeds unity”, our new data clearly exceeds unity gain (up to gain = 1.7, with negligible error bars), as shown in our new Fig. 4. We hope the reviewer will appreciate this improvement.

From the material science point of view, I tried to understand the origin of the negative differential resistance in LaCoO₃. Fig. 2c shows data only up to 4 mA of I_{bias} , which makes me wonder what is happening $I_{\text{bias}} > 4$ mA. I looked at Fig. S3 in the supplemental material, which shows even more puzzling data where we see an abrupt change in the I-V curve. In VO₂, similar data is obtained which are attributed to a Mott transition. In LaCoO₃, people found no feature of sudden transition as a function of temperature. Therefore, the sudden change in the I-V curve is interesting and puzzling to me.

I believe that material science is not the first priority of the paper, but the proof of principle for the active signal amplification is. However, their data is already interesting from a condensed matter physics point of view. I hope the authors address this issue before the publication of the paper.

The above point weakens their conclusion to me. Except for that, the paper is clearly written. The figures are also clear. The result and the logic are interesting. I thus recommend its publication only after they adequately addresses the issue.

We agree that the abrupt jump in the I-V curve is interesting from a physics point of view. We have performed additional work to clarify this issue.

In summary, (1) the jump is not an unknown phenomenon (we have added new references about it), (2) we understand the jump (we have new data for it), and (3) the jump does not affect our results (we have clarified this in our revision). We elaborate on all three points below.

-- The jump does not affect our results: In our devices, in some cases, the NDR occurs before the abrupt jump, and in other cases, after the jump; depending on the geometry of the device (e.g., comparing Supplementary Figs. XX and XX). In both cases, we show that edge of chaos or amplification corresponds to the NDR (and not to the jump). As such, the jump itself is not important to the demonstration here.

-- The jump is not an unknown phenomenon: We also note that the abrupt jump following the onset of NDR is not a new phenomenon. It was first observed and explained (with reasonably accuracy) in 1936 (Lueder, Schottky & Spenke, *Zur technischen Beherrschung des Wärmedurchschlags*).

Naturwissenschaften 24, 61 (1936)). Their measurement is reproduced below, which is similar to ours.

Fig. 1. Auftreten einer Abzweiggennlinie im Kennfeld einer Heißleiterscheibe.

Response Fig. 1: Fig. 1 reproduced from Naturwissenschaften 24, 61 (1936)

-- We understand the jump: The physics underlying this behavior corresponds to spatial localization of thermal distribution, which leads to minimization of energy (this has been hypothesized before). We have performed additional measurements on our devices (in-situ thermal and chemical mapping), along with modeling, which confirm this mechanism (thermal mapping reproduced below as an example).

Response Fig. 2: (a) Current-voltage behavior of a typical LaCoO_3 device. (b) In-situ thermal maps obtained during a current sweep, demonstrating strong localization of high-temperature regions concurrent with the abrupt jump in the voltage (red and purple arrows in panel (a)).

This kind of an interesting study deserves to be a separate and focused paper on its own, which is consistent with the reviewer's acknowledgement that the material science is not the first priority in the current paper. As such, we will not include many details of our new experiments in this paper, and instead, include them in a dedicated follow-on paper. We hope the reviewer understands and appreciates this choice.

Changes: Though we did not include all our new data in this paper, following the reviewer's suggestions, in the revision, we have provided sufficient detail to inform the reader about this behavior, with a brief overview of the underlying physics. Specifically, we have made the following changes (in the Methods section): (1) We have referenced the past papers that have explained this behavior, (2) we have provided a brief outline of the underlying localization associated with the abrupt change, and (3) we have outlined that the jump itself does not affect the measurements performed here (which are dependent on the NDR).

In addition, to aid researchers reproducing our measurements, we have displayed the quasi-static IV behavior (including the abrupt jump) for many devices with different geometries (Supplementary Figs. 5, 12, 14, 15).

Referee #3 (Remarks to the Author):

Active Signal Transmission at the Edge of Chaos

A. Summary of the key results

As a major novelty in science, the authors demonstrate how a small signal may be amplified over a certain frequency range as it travels across a purely-passive transmission line, if the latter is integrated on top of a substrate, composed of a new material, specifically LaCoO₃, biased on the negative differential resistance (NDR) region of its DC voltage (V) versus current (I) characteristic.

B. Originality and Significance

The paper reports a novel, original and highly-significant research work. It deserves publication on Nature in its current form.

Interestingly, a two-terminal memristor, composed of the new material, may be stabilized easily along the NDR region of its V-I locus, since, due to its mild conductivity versus temperature nonlinearity, is insensitive to the reactances in the DC measurement set-up. Moreover, when biased along the NDR region, its small-signal voltage and current signals appear to admit a relative phase difference ranging from $\pi/2$ to π in modulus. Using such a material to form an excitable substrate, lying under a purely-dissipative transmission line, the small signals respectively at the output and at the input of the transmission line itself were found to be shifted one relative to the other by at least $\pi/2$ and by at most π in modulus, while the output signal was found to be larger than the input signal across a certain frequency range, when the substrate was continuously polarized on the Edge of Chaos.

Very importantly, employing operando thermal mappings, it was further demonstrated how the energy, necessary for the amplification of the small signal across the transmission line, comes from the DC power supplied to the LaCoO₃ medium.

This remarkable research work may pave the way toward the future insertion of excitable media, biased on the Edge of Chaos, below lengthy on-chip transmission lines, to induce small signal amplification across them without the use of repeaters and buffers.

C. Data & methodology: validity of approach, quality of data, quality of presentation

This paper is an example of excellence in research.

It provides a comprehensive justification to the breakthrough research findings in the lab through a deep theoretical analysis corroborated by insightful model simulations.

It deserves to be published in Nature without further revision.

The authors of this manuscript include some of the most reliable and trustworthy device scientists on earth, and this adds credibility and solidity to the theoretico-experimental work.

Moreover, the paper is well written and the discussion of the research work is crystal clear.

Thank you! It is very rare to see such an effusively positive peer review, and we greatly appreciate your candor in expressing positive opinions. In addition, thanks for going through the manuscript with an eye for minute details. We truly appreciate the time and effort it must have taken you.

D. Appropriate use of statistics and treatment of uncertainties

Not applicable

E. Conclusions: robustness, validity, reliability

The authors have conducted a large number of additional experiments, besides those shown in the main manuscript, as reported in the supplementary information file, confirming the robustness of the phenomenal research findings.

F. Suggested improvements: experiments, data for possible revision

The following list of suggestions are very minor comments, which do not need another round of revision:

As a suggestion, it would be nice to add a short section, or even a brief remark, formalizing the difference between Edge of Chaos and semi-stable Edge of Chaos, already within the manuscript.

We have elaborated on this aspect in the last paragraph of the section ‘The criteria for semi-stable Edge of Chaos’ in the main text, as well as the first paragraph of the Methods (Nomenclature of EoC).

Caption of Fig. 1: negative differential behavior-> negative differential resistance behavior

We have made this correction.

Page 3 steady changes -> gentle changes

We have removed this phrase to enhance clarity.

Last line at page 5: (Fig. 3b) -> (Figs. 3b and 3c)

We have referenced both figure panels in this paragraph.

Fig. 3(c): replace π with $-\pi$ on the bottom of the color map

We have made this correction.

Fig. 3(c): add the indication of I_{bias} along the x axis, as done in Fig. 3(d)

We have made this correction.

Caption of Fig. 3(c): is displayed in Fig. 2d -> is displayed in Fig. 2c

We have made this correction.

In regard to the measurements on the circuit of Fig 3(a), it was said that, differently from what was done in the experiments on the test structure of Fig. 2(a), the voltage V_{bias} (v) served as the bias (small-signal) input variable, while the DC (small-signal) current I_{bias} (i) was employed as the output variable. Use one sentence to confirm whether the small-signal current i in the circuit of Fig. 3(a) was then converted back to some voltage v_{out} . This curiosity arises from the inspection of Fig. 4(a), referring however to another circuit set-up, where the output signal, specifically v_{out} , appears in voltage form.

We have clarified this issue with the following statement: “In the test structures (Fig. 2-3), we measured small-signal current outputs relative to small-signal voltage inputs, which allowed quantification of current-voltage phases. However, in the transmission lines, to measure gain across a common metric, we measured small-signal voltage output relative to small-signal voltage inputs.”

all three criteria three criteria-> all the three criteria

We have made this correction.

Page 8. Where it says “amplification decreased and fell below unity gain at about 4 kHz (Fig. 4c)” should 4 kHz rather be 10 kHz?

The reviewer is correct. We have revised the text.

Page 8. Where it says “as observed here (roughly 70 kHz)”, should it maybe be “as observed here (roughly 50 kHz)”, as mentioned some lines above?

The reviewer is correct. We have revised the text.

Caption of Fig. 4: “Normalized time evolution of input ($v_{in}(t)$) and output ($v_{out}(t)$) for two different I_{bias} – one in PDR (locally passive) and another in NDR (semi-stable EoC) of the corresponding DC response of the structure” -> Normalized time evolution of the input ($v_{in}(t)$) and output ($v_{out}(t)$) signals of the transmission line upon injecting either of two different currents (I_{bias}) into the LaCoO₃ substrate so as to poise it once in the PDR (locally passive) regime and once in the NDR (semi-stable EoC) regime.

We have made this change.

Page 12. It is said that the temporal data in Fig. 4b include the transmission line AC current responses to small signal stimuli under bias currents of 3mA and 4mA. **First of all, the AC responses are shown in voltage form and not in current form, right?** Secondly, did you really show the transmission line AC responses for both the bias current levels 3 mA and 4mA in Fig. 4b? Please check it.

The reference to Fig. 4b was a typographical error (it must have been Fig. 3b). With this correction, the text should make sense.

G. References: appropriate credit to previous work?

References to some other recent work on Edge of Chaos from the literature – e.g. the research studies on symmetry-breaking phenomena in homogeneous cellular media as well as on the impact of the Sodium (Na) ion channel on the emergence of the All-to-None complex neuronal phenomenon - could make the bibliography more comprehensive.

We have added relevant references (Refs. 25-26).

H. Clarity and context: lucidity of abstract/summary, appropriateness of abstract, introduction and conclusions

The paper is very clearly written. I believe the text and presentation of the work, including the illustrations, could not be done any better.

Reviewer Reports on the First Revision:

Referee #1 (Remarks to the Author):

I am very happy with the revision. The authors addressed my concerns adequately. The additional data makes the work convincing.

I would like to congratulate them on the work and am looking forward to seeing it published.

Referee #2 (Remarks to the Author):

I thank the authors for thoroughly answering my questions on the voltage jump following the negative differential resistance. I am satisfied with their response that the jump is not due to the material property of LaCoO_3 but a known phenomenon reported in 1939. The response Fig. 2 did convince me that temperature inhomogeneity is at the heart of the phenomenon. As stated in the previous review, this point does not affect the main arguments of the present paper. I don't have any more objections to the publication of this paper.

Referee #3 (Remarks to the Author):

I sincerely thank the authors for taking care of the few suggestions, which I collected in a short list of comments and submitted to their attention during the first review round.

Clearly, a multidisciplinary deep theoretical-experimental work, leveraging the strong technical know-how and complementary expertise of an excellent team of scientists, has been carried out for a long time to achieve the novel and exciting research findings communicated in a very clear yet comprehensive form in this paper. I confirm my strong recommendation for publishing the manuscript on Nature in its current version.